# Do Deep Neural Networks Suffer from Crowding?

**Anna Volokitin**[†♮]
voanna@vision.ee.ethz.ch

**Gemma Roig**[†‡ι]
gemmar@mit.edu

**Tomaso Poggio**[†‡]
tp@csail.mit.edu

[†]Center for Brains, Minds and Machines, Massachusetts Institute of Technology, Cambridge, MA
[‡]Istituto Italiano di Tecnologia at Massachusetts Institute of Technology, Cambridge, MA
[♮]Computer Vision Laboratory, ETH Zurich, Switzerland
[ι]Singapore University of Technology and Design, Singapore

## Abstract

Crowding is a visual effect suffered by humans, in which an object that can be recognized in isolation can no longer be recognized when other objects, called flankers, are placed close to it. In this work, we study the effect of crowding in artificial Deep Neural Networks (DNNs) for object recognition. We analyze both deep convolutional neural networks (DCNNs) as well as an extension of DCNNs that are multi-scale and that change the receptive field size of the convolution filters with their position in the image. The latter networks, that we call eccentricity-dependent, have been proposed for modeling the feedforward path of the primate visual cortex. Our results reveal that the eccentricity-dependent model, trained on target objects in isolation, can recognize such targets in the presence of flankers, if the targets are near the center of the image, whereas DCNNs cannot. Also, for all tested networks, when trained on targets in isolation, we find that recognition accuracy of the networks decreases the closer the flankers are to the target and the more flankers there are. We find that visual similarity between the target and flankers also plays a role and that pooling in early layers of the network leads to more crowding. Additionally, we show that incorporating flankers into the images of the training set for learning the DNNs does not lead to robustness against configurations not seen at training.

## 1  Introduction

Despite stunning successes in many computer vision problems [1, 2, 3, 4, 5], Deep Neural Networks (DNNs) lack interpretability in terms of how the networks make predictions, as well as how an arbitrary transformation of the input, such as addition of clutter in images in an object recognition task, will affect the function value.

Examples of an empirical approach to this problem are testing the network with adversarial examples [6, 7] or images with different geometrical transformations such as scale, position and rotation, as well as occlusion [8]. In this paper, we add clutter to images to analyze the crowding in DNNs.

Crowding is a well known effect in human vision [9, 10], in which objects (targets) that can be recognized in isolation can no longer be recognized in the presence of nearby objects (flankers), even though there is no occlusion. We believe that crowding is a special case of the problem of clutter in object recognition. In crowding studies, human subjects are asked to fixate at a cross at the center of a screen, and objects are presented at the periphery of their visual field in a flash such that the subject has no time to move their eyes. Experimental data suggests that crowding depends on the distance of the target and the flankers [11], eccentricity (the distance of the target to the fixation point), as well as the similarity between the target and the flankers [12, 13] or the configuration of the flankers around the target object [11, 14, 15].

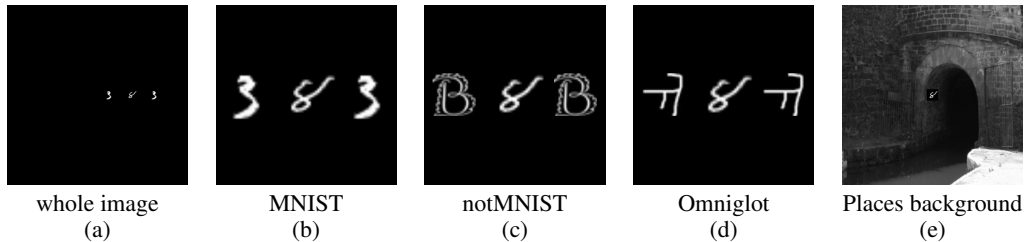

| whole image | MNIST | notMNIST | Omniglot | Places background |
|:---:|:---:|:---:|:---:|:---:|
| (a) | (b) | (c) | (d) | (e) |

Figure 1: (a) Example image used to test the models, with even MNIST as target and two odd MNIST flankers. (b-d) Close-up views with odd MNIST, notMNIST and Omniglot datasets as flankers, respectively. (e) An even MNIST target embedded into a natural image.

Many computational models of crowding have been proposed *e.g.* [16, 17]. Our aim is not to model human crowding. Instead, we characterize the crowding effect in DNNs trained for object recognition, and analyze which models and settings suffer less from such effects.

We investigate two types of DNNs for crowding: traditional deep convolutional neural networks and an extension of these which is multi-scale, and called eccentricity-dependent model [18]. Inspired by the retina, the receptive field size of the convolutional filters in this model grows with increasing distance from the center of the image, called the eccentricity. Cheung *et al.* [19] explored the emergence of such property when the visual system has an eye-fixation mechanism.

We investigate under which conditions crowding occurs in DNNs that have been trained with images of target objects in isolation. We test the DNNs with images that contain the target object as well as clutter, which the network has never seen at training. Examples of the generated images using MNIST [20], notMNIST [21], and Omniglot [22] datasets are depicted in Fig 1, in which even MNIST digits are the target objects. As done in human psychophysics studies, we take recognition accuracy to be the measure of crowding. If a DNN can recognize a target object correctly despite the presence of clutter, crowding has not occurred.

Our experiments reveal the dependence of crowding on image factors, such as flanker configuration, target-flanker similarity, and target eccentricity. Our results also show that prematurely pooling signals increases crowding. This result is related to the theories of crowding in humans. In addition, we show that training the models with cluttered images does not make models robust to clutter and flankers configurations not seen in training. Thus, training a model to be robust to general clutter is prohibitively expensive.

We also discover that the eccentricity-dependent model, trained on isolated targets, can recognize objects even in very complex clutter, *i.e.* when they are embedded into images of places (Fig 1(e)). Thus, if such models are coupled with a mechanism for selecting eye fixation locations, they can be trained with objects in isolation being robust to clutter, reducing the amount of training data needed.

## 2 Models

In this section we describe the DNN architectures for which we characterize crowding effect. We consider two kinds of DNN models: Deep Convolutional Neural Networks and eccentricity-dependent networks, each with different pooling strategies across space and scale. We investigate pooling in particular, because we [18, 23] as well as others [24] have suggested that feature integration by pooling may be the cause of crowding in human perception.

### 2.1 Deep Convolutional Neural Networks

The first set of models we investigate are deep convolutional neural networks (DCNN) [25], in which the image is processed by three rounds of convolution and max pooling across space, and then passed to one fully connected layer for the classification. We investigate crowding under three different spatial pooling configurations, listed below and shown in Fig 2. The word pooling in the names of the model architectures below refers to how quickly we decrease the feature map size in the model. All architectures have $3 \times 3$ max pooling with various strides, and are:

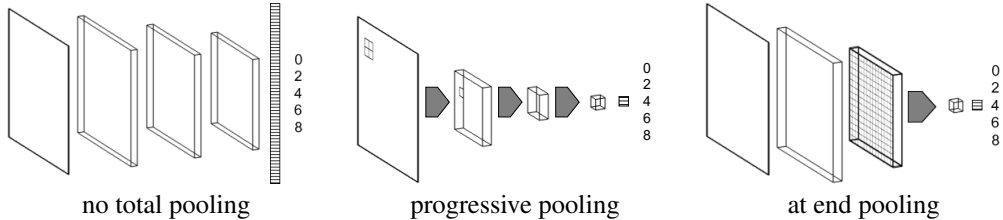

no total pooling        progressive pooling        at end pooling

Figure 2: DCNN architectures with three convolutional layers and one fully connected, trained to recognize even MNIST digits. These are used to investigate the role of pooling in crowding. The grey arrow indicates downsampling.

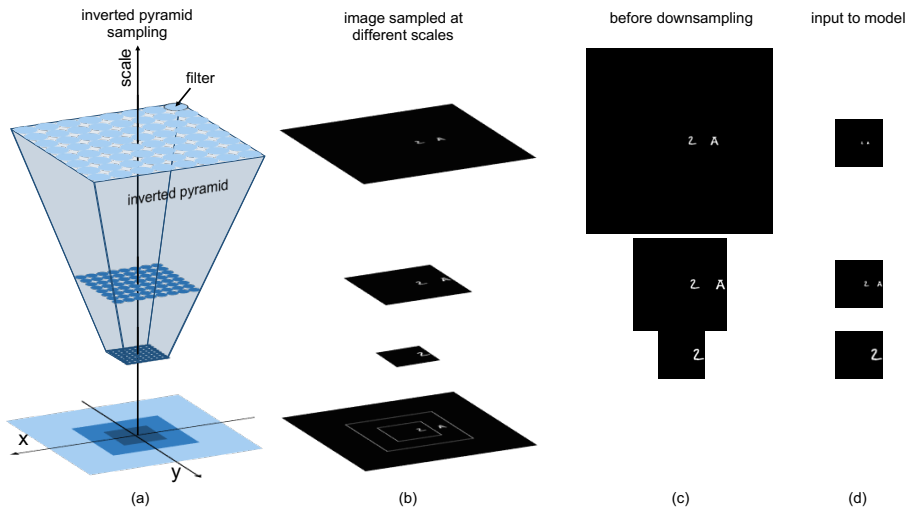

Figure 3: Eccentricity-dependent model: Inverted pyramid with sampling points. Each circle represents a filter with its respective receptive field. For simplicity, the model is shown with 3 scales.

- **No total pooling** Feature maps sizes decrease only due to boundary effects, as the 3×3 max pooling has stride 1. The square feature maps sizes after each pool layer are 60-54-48-42.
- **Progressive pooling** 3×3 pooling with a stride of 2 halves the square size of the feature maps, until we pool over what remains in the final layer, getting rid of any spatial information before the fully connected layer. (60-27-11-1).
- **At end pooling** Same as *no total pooling*, but before the fully connected layer, max-pool over the entire feature map. (60-54-48-1).

The data in each layer in our model is a 5-dimensional tensor of `minibatch size`× x × y × `number of channels`, in which x defines the width and y the height of the input. The input image to the model is resized to $60 \times 60$ pixels. In our training, we used minibatches of 128 images, 32 feature channels for all convolutional layers, and convolutional filters of size $5 \times 5$ and stride $1$.

## 2.2 Eccentricity-dependent Model

The second type of DNN model we consider is an eccentricity-dependent deep neural network, proposed by Poggio *et al.* in [18] as a model of the human visual cortex and further studied in [23]. Its eccentricity dependence is based on the human retina, which has receptive fields which increase in size with eccentricity. [18] argues that the computational reason for this property is the need to compute a scale- and translation-invariant representation of objects. [18] conjectures that this model is robust to clutter when the target is near the fixation point.

As discussed in [18], the set of all scales and translations for which invariant representations can be computed lie within an inverted truncated pyramid shape, as shown in Fig 3(a). The width of the pyramid at a particular scale is roughly related to the amount of translation invariance for objects of that size. Scale invariance is prioritized over translation invariance in this model, in contrast to

classical DCNNs. From a biological point of view, the limitation of translation invariance can be compensated for by eye movements, whereas to compensate for a lack of scale invariance the human would have to move their entire body to change their distance to the object.

The eccentricity-dependent model computes an invariant representation by sampling the inverted pyramid at a discrete set of scales with the same number of filters at each scale. At larger scales, the receptive fields of the filters are also larger to cover a larger image area, see Fig 3(a). Thus, the model constructs a multi-scale representation of the input, where smaller sections (crops) of the image are sampled densely at a high resolution, and larger sections (crops) are sampled with at a lower resolution, with each scale represented using the same number of pixels, as shown in Fig 3(b-d). Each scale is treated as an input channel to the network and then processed by convolutional filters, the weights of which are shared also across scales as well as space. Because of the downsampling of the input image, this is equivalent to having receptive fields of varying sizes. These shared parameters also allow the model to learn a scale invariant representation of the image.

Each processing step in this model consists of convolution-pooling, as above, as well as max pooling across different scales. Scale pooling reduces the number of scales by taking the maximum value of corresponding locations in the feature maps across multiple scales. We set the spatial pooling constant using *At end pooling*, as described above. The type of scale pooling is indicated by writing the number of scales remaining in each layer, *e.g. 11-1-1-1-1*. The three configurations tested for scale pooling are (1) **at the beginning**, in which all the different scales are pooled together after the first layer, *11-1-1-1-1* (2) **progressively**, *11-7-5-3-1* and (3) **at the end**, *11-11-11-11-1*, in which all 11 scales are pooled together at the last layer.

The parameters of this model are the same as for the DCNN explained above, except that now there are extra filters for the scales. Note that because of weight sharing across scales, the number of parameters in the eccentricity dependent model is equal that in a standard DCNN. We use 11 crops, with the smallest crop of $60 \times 60$ pixels, increasing by a factor of $\sqrt{2}$. Exponentially interpolated crops produce fewer boundary effects than linearly interpolated crops, while having qualitatively the same behavior. Results with linearly extracted crops are shown in Fig 7 of the supplementary material. All the crops are resized to $60 \times 60$ pixels, which is the same input image size used for the DCNN above. Image crops are shown in Fig 9.

**Contrast Normalization** We also investigate the effect of input normalization so that the sum of the pixel intensities in each scale is in the same range. To de-emphasize the smaller crops, which will have the most non-black pixels and therefore dominate the max-pooling across scales, in some experiments we rescale all the pixel intensities to the [0, 1] interval, and then divide them by factor proportional to the crop area $((\sqrt{2})^{11-i}$, where $i = 1$ for the smallest crop).

## 3 Experimental Set-up

Models are trained with back-propagation to recognize a set of objects, which we call *targets*. During testing, we present the models with images which contain a target object as well as other objects which the model has not been trained to recognize, which we call *flankers*. The flanker acts as clutter with respect to the target object.

Specifically, we train our models to recognize even MNIST digits—*i.e.* numbers $0, 2, 4, 6, 8$—shifted at different locations of the image along the horizontal axis, which are the target objects in our experiments. We compare performance when we use images with the target object in isolation, or when flankers are also embedded in the training images. The flankers are selected from odd MNIST digits, notMNIST dataset [21] which contains letters of different typefaces, and Omniglot [22] which was introduced for one-shot character recognition. Also, we evaluate recognition when the target is embedded to images of the Places dataset [26].

The images are of size 1920 squared pixels, in which we embedded target objects of 120 squared px, and flankers of the same size, unless contrary stated. Recall that the images are resized to $60 \times 60$ as input to the networks. We keep the training and testing splits provided by the MNIST dataset, and use it respectively for training and testing. We illustrate some examples of target and flanker configuration in Fig 1. We refer to the target as *a* and to the flanker as *x* and use this shorthand in the plots. All experiments are done in the right half of the image plane. We do this to check if there is a difference between central and peripheral flankers. We test the models under 4 conditions:

- *No flankers.* Only the target object. (*a* in the plots)
- *One central flanker* closer to the center of the image than the target. (*xa*)
- *One peripheral flanker* closer to the boundary of the image that the target. (*ax*)
- *Two flankers* spaced equally around the target, being both the same object, see Fig 1. (*xax*).

## 4  Experiments

In this section, we investigate the crowding effect in DNNs. We first carry out experiments on models that have been trained with images containing both targets and flankers. We then repeat our analysis with the models trained with images of the targets in isolation, shifted at all positions in the horizontal axis. We analyze the effect of flanker configuration, flanker dataset, pooling in the model architecture, and model type, by evaluating accuracy recognition of the target objects.[1]

### 4.1  DNNs Trained with Target and Flankers

In this setup we trained DNNs with images in which there were two identical flankers randomly chosen from the training set of MNIST odd digits, placed at a distance of 120 pixels on either side of the target (*xax*). The target is shifted horizontally, while keeping the distance between target and flankers constant, called the *constant spacing* setup, and depicted in Fig 1(a) of the supplementary material. We evaluate (i) DCNN with at the end pooling, and (ii) eccentricity-dependent model with *11-11-11-11-1* scale pooling, at the end spatial pooling and contrast normalization. We report the results using the different flanker types at test with *xax*, *ax*, *xa* and *a* target flanker configuration, in which *a* represents the target and *x* the flanker, as described in Section 3. [2]

Results are in Fig 4. In the plots with 120 px spacing, we see that the models are better at recognizing objects in clutter than isolated objects for all image locations tested, especially when the configuration of target and flanker is the same at the training images than in the testing images (*xax*). However, in the plots where target-flanker spacing is 240 px recognition accuracy falls to less than the accuracy of recognizing isolated target objects. Thus, in order for a model to be robust to all kinds of clutter, it needs to be trained with all possible target-flanker configurations, which is infeasible in practice.

Interestingly, we see that the eccentricity model is much better at recognizing objects in isolation than the DCNN. This is because the multi-scale crops divide the image into discrete regions, letting the model learn from image parts as well as the whole image.

We performed an additional experiment training the network with images that contain the same target-flanker configuration as above *(xax)*, but with different spacings between the target and the flankers, including different spacings on either side of the target. Left spacing and right spacing are sampled from 120 px, 240 px and 480 px independently. Train and test images shown in Fig. 2 of supplementary material.

We test two conditions: (1) With flankers on both sides from the target (*xax*) at a spacing not seen in the training set (360 px); (2) With 360 px spacing, including 2 flankers on both sides (4 flankers total, *xxaxx*). In Fig 5, we show the accuracy for images with 360 px target-flanker spacing, and see that accuracy is not impaired, neither for the DCNN nor the eccentricity model. Yet, the DCNN accuracy for images with four flankers is impaired, while the eccentricity model still has unimpaired recognition accuracy provided that the target is in the center of the image.

Thus, the recognition accuracy is not impaired for all tested models when flankers are in a similar configuration in testing as in training. This is even when the flankers at testing are placed at a spacing that is in between two seen spacings used at training. The models can interpolate to new spacings of flankers when using similar configurations in test images as seen during training, *e.g. (xax)*, arguably due to the pooling operators. Yet, DCNN recognition is still severely impaired and do not generalize to new flanker configurations, such as adding more flankers, when there are 2 flankers on both sides of the target *(xxaxx)*. To gain robustness to such configurations, each of these cases should be explicitly included in the training set. Only the eccentricity-dependent model is robust to different flanker

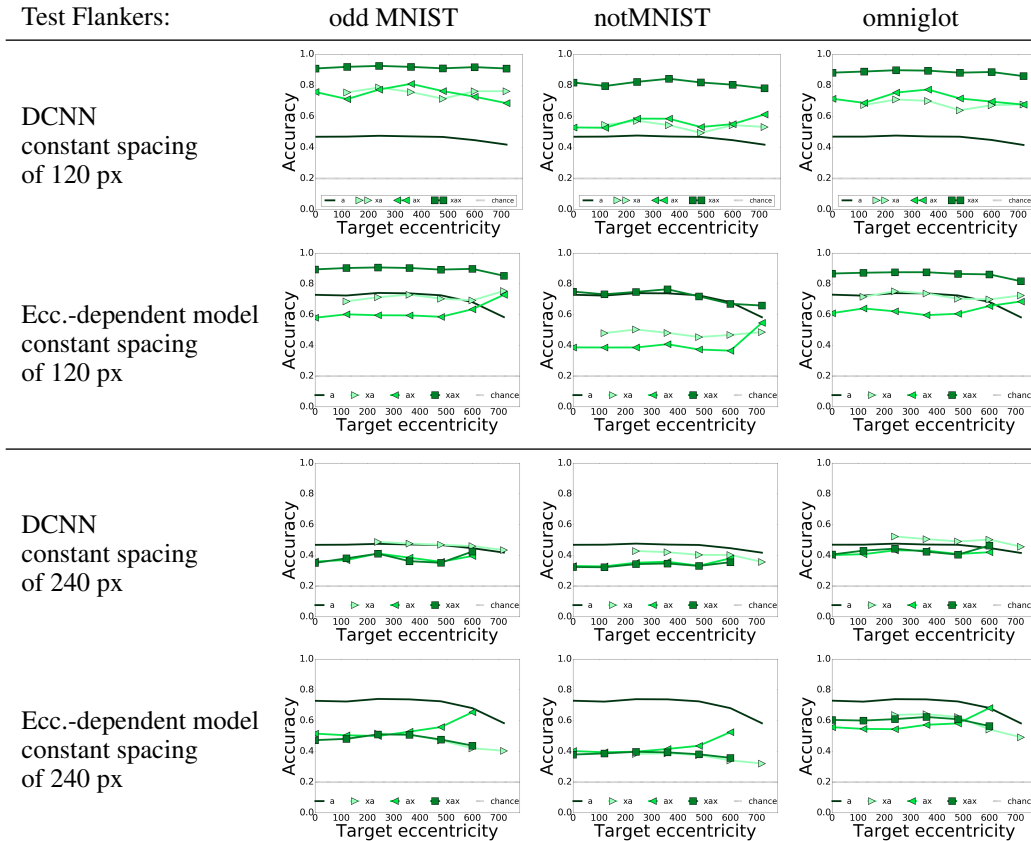

Figure 4: Even MNIST accuracy recognition of DCNN (at the end pooling) and Eccentricity Model (11-11-11-11-1, At End spatial pooling with contrast normalization) trained with odd MNIST flankers at 120px constant spacing. The target eccentricity is in pixels.

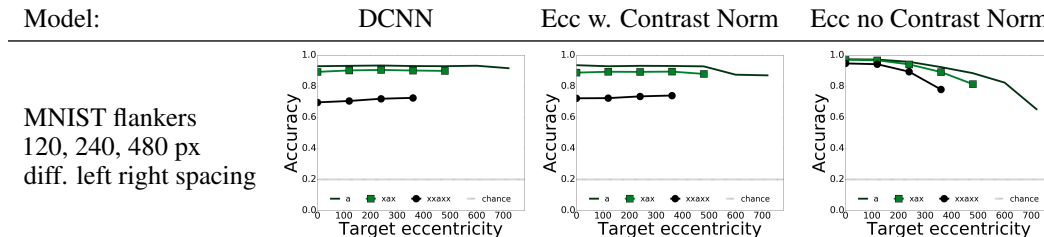

Figure 5: All models tested at 360 px target-flanker spacing. All models can recognize digit in the presence of clutter at a spacing that is in between spacings seen at training time. However, the eccentricity Model (11-11-11-11-1, At End spatial pooling with contrast normalization) and the DCNN fail to generalize to new types of flanker configurations (two flankers on each side, *xxaxx*) at 360 px spacing between the target and inner flanker

configurations not included in training, when the target is centered. We will explore the role of contrast normalization in Sec 4.3.

## 4.2 DNNs Trained with Images with the Target in Isolation

For these experiments, we train the models with the target object in isolation and in different positions of the image horizontal axis. We test the models on images with target-flanker configurations *a*, *ax*, *xa*, *xax*.

**DCNN** We examine the crowding effect with different spatial pooling in the DCNN hierarchy: (i) no total pooling, (ii) progressive pooling, (iii) at end pooling (see Section 2.1 and Fig 2).

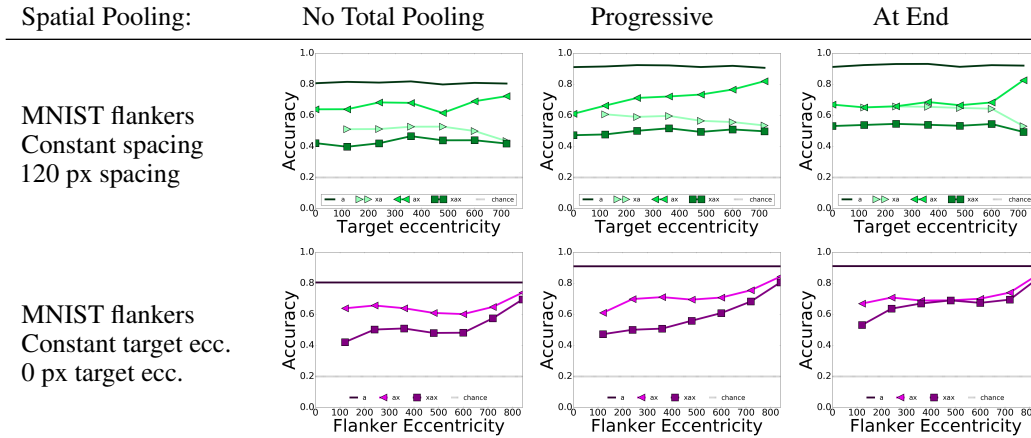

| Spatial Pooling: | No Total Pooling | Progressive | At End |
| --- | --- | --- | --- |

Figure 6: Accuracy results of 4 layer DCNN with different pooling schemes trained with targets shifted across image and tested with different flanker configurations. Eccentricity is in pixels.

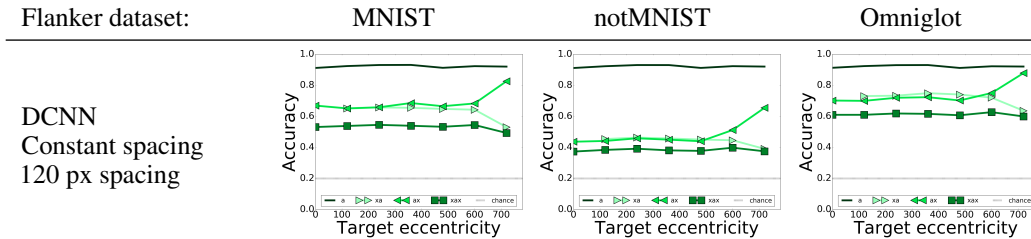

| Flanker dataset: | MNIST | notMNIST | Omniglot |
| --- | --- | --- | --- |

Figure 7: Effect in the accuracy recognition in DCNN with *at end pooling*, when using different flanker datasets at testing.

Results are shown in Fig 6. In addition to the *constant spacing* experiment (see Section 4.1), we also evaluate the models in a setup called *constant target eccentricity*, in which we have fixed the target in the center of the visual field, and change the spacing between the target and the flanker, as shown in Fig 1(b) of the supplementary material. Since the target is already at the center of the visual field, a flanker can not be more central in the image than the target. Thus, we only show *x, ax* and *xax* conditions.

From Fig 6, we observe that the more flankers are present in the test image, the worse recognition gets. In the constant spacing plots, we see that recognition accuracy does not change with eccentricity, which is expected, as translation invariance is built into the structure of convolutional networks. We attribute the difference between the *ax* and *xa* conditions to boundary effects. Results for notMNIST and Omniglot flankers are shown in Fig 4 of the supplementary material.

From the constant target eccentricity plots, we see that as the distance between target and flanker increases, the better recognition gets. This is mainly due to the pooling operation that merges the neighboring input signals. Results with the target at the image boundary is shown in Fig 3 of the supplementary material.

Furthermore, we see that the network called *no total pooling* performs worse in the no flanker setup than the other two models. We believe that this is because pooling across spatial locations helps the network learn invariance. However, in the below experiments, we will see that there is also a limit to how much pooling across scales of the eccentricity model improves performance.

We test the effect of flankers from different datasets evaluating DCNN model with *at end pooling* in Fig 7. Omniglot flankers crowd slightly less than odd MNIST flankers. The more similar the flankers are to the target object—even MNIST, the more recognition impairment they produce. Since Omniglot flankers are visually similar to MNIST digits, but not digits, we see that they activate the convolutional filters of the model less than MNIST digits, and hence impair recognition less.

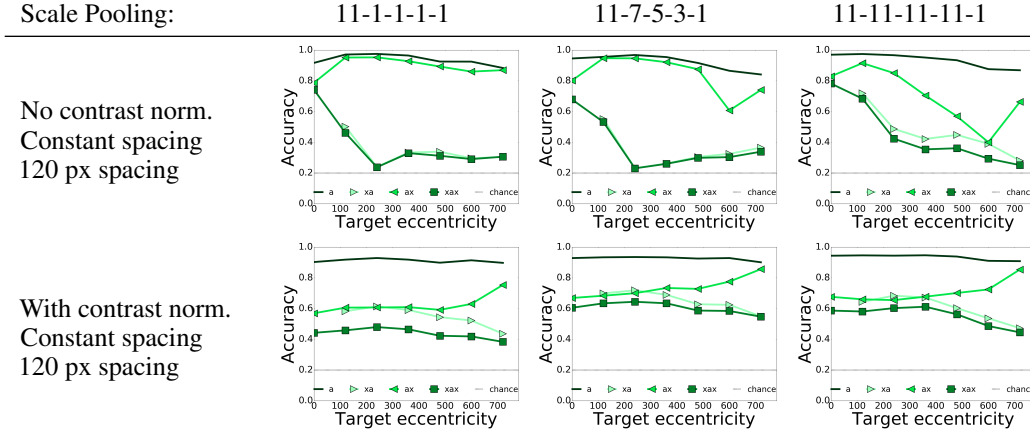

Figure 8: Accuracy performance of Eccentricity-dependent model with spatial *At End* pooling, and changing contrast normalization and scale pooling strategies. Flankers are odd MNIST digits.

We also observe that notMNIST flankers crowd much more than either MNIST or Omniglot flankers, even though notMNIST characters are much more different to MNIST digits than Omniglot flankers. This is because notMNIST is sampled from special font characters and these have many more edges and white image pixels than handwritten characters. In fact, both MNIST and Omniglot have about 20% white pixels in the image, while notMNIST has 40%. Fig 5 of the supplementary material shows the histogram of the flanker image intensities. The high number of edges in the notMNIST dataset has a higher probability of activating the convolutional filters and thus influencing the final classification decision more, leading to more crowding.

**Eccentricity Model** We now repeat the above experiment with different configurations of the eccentricity dependent model. In this experiment, we choose to keep the spacial pooling constant (*at end pooling*), and investigate the effect of pooling across scales, as described in Section 2.2. The three configurations for scale pooling are (1) at the beginning, (2) progressively and (3) at the end. The numbers indicate the number of scales at each layer, so 11-11-11-11-1 is a network in which all 11 scales have been pooled together at the last layer.

Results with odd MNIST flankers are shown in Fig 8. Our conclusions for the effect of the flanker dataset are similar to the experiment above with DCNN. (Results with other flanker datasets shown in Fig 6 of the supplementary material.)

In this experiment, there is a dependence of accuracy on target eccentricity. The model without contrast normalization is robust to clutter at the fovea, but cannot recognize cluttered objects in the periphery. Interestingly, also in psychophysics experiments little effect of crowding is observed at the fovea [10]. The effect of adding one central flanker (*ax*) is the same as adding two flankers on either side (*xax*). This is because the highest resolution area in this model is in the center, so this part of the image contributes more to the classification decision. If a flanker is placed there instead of a target, the model tries to classify the flanker, and, it being an unfamiliar object, fails. The dependence of accuracy on eccentricity can however be mitigated by applying contrast normalization. In this case, all scales contribute equally in contrast, and dependence of accuracy on eccentricity is removed.

Finally, we see that if scale pooling happens too early in the model, such as in the 11-1-1-1-1 architecture, there is more crowding. Thus, pooling too early in the architecture prevents useful information from being propagated to later processing in the network. For the rest of the experiments, we always use the 11-11-11-11-1 configuration of the model with spatial pooling at the end.

## 4.3 Complex Clutter

Previous experiments show that training with clutter does not give robustness to clutter not seen in training, *e.g.* more or less flankers, or different spacing. Also, that the eccentricity-dependent model is more robust to clutter when the target is at the image center and no contrast normalization is applied, Fig 8. To further analyze the models robustness to other kinds of clutter, we test models trained

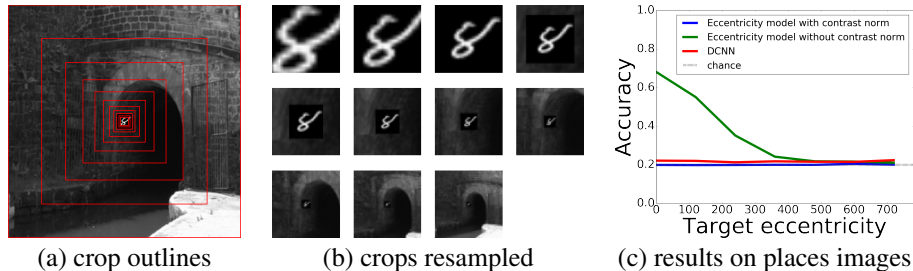

|   (a) crop outlines   |   (b) crops resampled   |   (c) results on places images   |

Figure 9: (a-b) An example of how multiple crops of an input image look, as well as (c) recognition accuracy when MNIST targets are embedded into images of places.

on images with isolated targets shifted along the horizontal axis, with images in which the target is embedded into randomly selected images of Places dataset [26], shown in Fig 1(e) and Fig 9(a), (b).

We tested the DCNN and the eccentricity model (11-11-11-11-1) with and without contrast normalization, both with *at end pooling*. The results are in Fig 9(c): only the eccentricity model without contrast normalization can recognize the target and only when the target is close to the image center. This implies that the eccentricity model is robust to clutter: it doesn't need to be trained with all different kinds of clutter. If it can fixate on the relevant part of the image, it can still discriminate the object, even at multiple object scales because this model is scale invariant [18].

## 5   Discussion

We investigated whether DNNs suffer from crowding, and if so, under which conditions, and what can be done to reduce the effect. We found that DNNs suffer from crowding. We also explored the most obvious approach to mitigate this problem, by including clutter in the training set of the model. Yet, this approach does not help recognition in crowding, unless, of course, a similar configuration of clutter is used for training and testing.

We explored conditions under which DNNs trained with images of targets in isolation are robust to clutter. We trained various architectures of both DCNNs and eccentricity-dependent models with images of isolated targets, and tested them with images containing a target at varying image locations and 0, 1 or 2 flankers, as well as with the target object embedded into complex scenes. We found the four following factors influenced the amount of crowding in the models:

- **Flanker Configuration:** When models are trained with images of objects in isolation, adding flankers harms recognition. Adding two flankers is the same or worse than adding just one and the smaller the spacing between flanker and target, the more crowding occurs. These is because the pooling operation merges nearby responses, such as the target and flankers if they are close.
- **Similarity between target and flanker:** Flankers more similar to targets cause more crowding, because of the selectivity property of the learned DNN filters.
- **Dependence on target location and contrast normalization:** In DCNNs and eccentricity-dependent models with contrast normalization, recognition accuracy is the same across all eccentricities. In eccentricity-dependent networks without contrast normalization, recognition does not decrease despite presence of clutter when the target is at the center of the image.
- **Effect of pooling:** adding pooling leads to better recognition accuracy of the models. Yet, in the eccentricity model, pooling across the scales too early in the hierarchy leads to lower accuracy.

Our main conclusion is that when testing accuracy recognition of the target embedded in (place) images, the eccentricity-dependent model – without contrast normalization and with spatial and scale pooling at the end of the hierarchy – is robust to complex types of clutter, even though it had been trained on images of objects in isolation. Yet, this occurs only when the target is at the center of the image as it occurs when it is fixated by a human observer. Our analysis suggests that if we had access to a system for selecting target object location, such as the one proposed by [27], the eccentricity dependent model could be trained with lower sample complexity than other DCNN because it is robust to some factors of image variation, such as clutter and scale changes. Translation invariance would mostly be achieved through foveation.

**Acknowledgments**

This work was supported by the Center for Brains, Minds and Machines (CBMM), funded by NSF STC award CCF - 1231216. A. Volokitin was also funded by Swiss Commission for Technology and Innovation (KTI, Grant No 2-69723-16), and thanks Luc Van Gool for his support. G. Roig was partly funded by SUTD SRG grant (SRG ISTD 2017 131). We also thank Xavier Boix, Francis Chen and Yena Han for helpful discussions.

## Footnotes

[1]Code to reproduce experiments is available at `https://github.com/CBMM/eccentricity`

[2]The *ax* flanker line starts at 120 px of target eccentricity, because nothing was put at negative eccentricities. For the case of 2 flankers, when the target was at 0-the image center, the flankers were put at -120 px and 120 px.

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
