[Supplementary Material]

# Do Deep Neural Networks Suffer from Crowding? Supplementary Material

**Anna Volokitin**[†♮]
voanna@vision.ee.ethz.ch

**Gemma Roig**[†‡ι]
gemmar@mit.edu

**Tomaso Poggio**[†‡]
tp@csail.mit.edu

[†]Center for Brains, Minds and Machines, Massachusetts Institute of Technology, Cambridge, MA
[‡]Istituto Italiano di Tecnologia at Massachusetts Institute of Technology, Cambridge, MA
[♮]Computer Vision Laboratory, ETH Zurich, Switzerland
[ι]Singapore University of Technology and Design, Singapore

Here, we report the complementary results that were left for the supplementary material in the main paper.

## 1  Experiments

To investigate the crowding effect in DNNs, we change the target-flanker configuration in two ways. In one case, we always place the flanker at a fixed distance from the target, and then change the target eccentricity. We call this the *constant spacing* setup, shown in Fig. 1(a). In second case, we place the target at a certain eccentricity and increase the target-flanker spacing, as shown in Fig. 1(b).

(a) constant spacing between target and flanker, but varying eccentricities

(b) target is at constant eccentricity, but we change the spacing

Figure 1: Examples of input images used in the *constant spacing* and *constant target* experiments.

|  (a) | (b) | (c) | (d) |

Figure 2: (a) and (b) show images used to train the model for the interpolation experiment in Section 4.1. Left and right spacings are independently sampled from 120, 240, 480 px. (c) is an example test image with a target-flanker spacing not seen in training, namely 360px. (d) is an example of an image with a flanker configuration not seen in training at all, with double flankers on each side.

| Spatial Pooling: | No Total Pooling | Progressive | At End |

MNIST flankers
Constant target ecc.
720 px target ecc.

Figure 3: Accuracy results of 4 layer DCNN with different pooling schemes trained with targets shifted across image and tested with different flanker configurations. Eccentricity is in pixels.

## 1.1 DNNs Trained with Target and Flankers

We performed an experiment training the network with images that contain a target-flanker configuration with different spacings between the target and the flankers, including different spacings on either side of the target. Left spacing and right spacing are sampled from 120 px, 240 px and 480 px independently, as shown in Fig. 2a,b. We display an example of a test image with a target-flanker spacing not seen in training, namely with flankers at 360px spacing, in Fig. 2c. In Fig. 2d, we show an example of an image with a flanker configuration not seen in training, with double flankers on each side.

## 1.2 DNNs Trained with Images with the Target in Isolation

For these experiments, we train the models with the target object in isolation and in different positions of the image horizontal axis. We test the models on images with target-flanker configurations *a*, *ax*, *xa*, *xax*.[1]

**DCNN**   We examine the crowding effect with different spatial pooling in the DCNN hierarchy: (i) no total pooling, (ii) progressive pooling, (iii) at end pooling (explained in the main paper).

In addition to the evaluation of DCNNs in *constant target eccentricity* at 240 pixels, reported in the main paper, here, we test them with images in which we have fixed the target at 720 pixels from the center of the image, as shown in Fig 3. Since the target is already at the edge of the visual field, a flanker can not be more peripheral in the image than the target. Thus, we only show *x* and *xa* conditions. Same results as for the 240 pixels target eccentricity can be extracted. The closer the flanker is to the target, the more accuracy decreases. Also, we see that when the target is so close to the image boundary, recognition is poor because of boundary effects eroding away information about the target.

We also show the *constant spacing* results in Fig 4 for experiments with flankers from the notMNIST and Omniglot datasets, in addition to the results with odd MNIST flankers from the main paper. Here again we observe that the more flankers are present in the test image, the worse recognition gets. Here, we see that recognition accuracy does not change with eccentricity, which is expected, as translation invariance is built into the structure of convolutional networks. We attribute the difference between the *ax* and *xa* conditions to boundary effects. Here we see that Omniglot flankers crowd slightly less than odd MNIST flankers. The more similar the flankers are to the target object—even MNIST, the more recognition impairment they produce. Since

| | Spatial Pooling | | |
| | No Total Pooling | Progressive | At End |
| Constant spacing notMNIST flankers | | | |
| Constant spacing Omniglot flankers | | | |

Figure 4: Accuracy results of 4 layer DCNN with different pooling schemes trained with targets shifted across image and tested with different flanker configurations.

Figure 5: Histograms for relative average pixel intensities of the flanker datasets. 10000 images were sampled randomly from each set.

Omniglot flankers are visually similar to MNIST digits, but not digits, we see that they activate the convolutional filters of the model less than MNIST digits, and hence impair recognition less.

We also observe that notMNIST flankers crowd much more than either Omniglot flankers, even though notMNIST characters are much more different to MNIST digits than Omniglot flankers. This is because notMNIST is sampled from special font characters and these have many more edges and white image pixels than handwritten characters. In fact, both MNIST and Omniglot have about 20% white pixels in the image, while notMNIST has 40%. A histogram of relative image image intensities is shown in Fig 5. The high number of edges in the notMNIST dataset has a higher probability of activating the convolutional filters and thus influencing the final classification decision more, leading to more crowding.

**Eccentricity Model** We now repeat the above experiment with different configurations of the eccentricity dependent model. We choose to keep the spacial pooling constant (*at end pooling*), and investigate the effect of pooling across scales. The three configurations for scale pooling are (1) at the beginning, (2) progressively and (3) at the end. The numbers indicate the number of scales at each layer, so 11-11-11-11-1 is a network in which all 11 scales have been pooled together at the last layer.

We reported results with odd MNIST flankers in the main paper. Results with notMNIST and Omniglot flankers are shown in Fig 6. Our conclusions for the effect of the flanker dataset are similar to the experiment in the main paper with odd MNIST flankers, which we summarize next.

In Fig. 6, we observe that there is a dependence of accuracy on target eccentricity. The model without contrast normalization is robust to clutter at when it is placed at the image center, but cannot recognize cluttered objects in the periphery. The effect of adding one central flanker (*ax*) is the same as adding two flankers on either side (*xax*). This is because the highest resolution area in our model is in the center, so this part of the image contributes more to the classification decision. If a flanker is placed there instead of a target, the model tries to classify the flanker, and, it being an unfamiliar object, fails.

The dependence of eccentricity in accuracy can however be mitigated by applying contrast normalization. In this case, all scales contribute equally in contrast, and dependence of accuracy on eccentricity is removed.

Figure 6: Accuracy performance of eccentricity-dependent model with spatial *At End pooling*, and changing contrast normalization and scale pooling strategies. Flankers are from notMNIST and Omniglot datasets.

Finally, we see that if scale pooling happens too early in the model, such as in the 11-1-1-1-1 architecture, there is more crowding. Thus, pooling too early in the architecture prevents useful information from being propagated to later processing in the network. For the rest of the experiments, we always use the 11-11-11-11-1 configuration of the model with spatial pooling at the end.

Finally, we also show the results of this experiment with a eccentricity-dependent model with crops that are linearly interpolated. As in the exponential interpolation case reported in the main paper, we use 11 crops, with the smallest crop of $60 \times 60$ pixels, increasing by a linear factor up to the image size (1920 squared pixels). All the crops are resized to $60 \times 60$ pixels as in the crops exponential interpolation case. In in Fig. 7 we see that the conclusions are the same as for the exponentially interpolated crops, yet there are more boundary effects in the linearly interpolated crops, while having qualitatively the same behavior.

### Acknowledgments

This work was supported by the Center for Brains, Minds and Machines (CBMM), funded by NSF STC award CCF - 1231216. A. Volokitin was also funded by Swiss Commission for Technology and Innovation (KTI, Grant No 2-69723-16), and thanks Luc Van Gool for his support. G. Roig was partly funded by SUTD SRG grant (SRG ISTD 2017 131). We also thank Xavier Boix, Francis Chen and Yena Han for helpful discussions.

| Scale Pooling: | 11-1-1-1-1 | 11-7-5-3-1 | 11-11-11-11-1 |
|---|---|---|---|
| Linear crops<br>No contrast norm.<br>MNIST flankers<br>Constant spacing<br>120 px spacing | | | |
| Linear crops<br>With contrast norm.<br>MNIST flankers<br>Constant spacing<br>120 px spacing | | | |

Figure 7: Accuracy performance of eccentricity-dependent model with linearly interpolated crops with spatial *At End pooling*, and changing contrast normalization and scale pooling strategies. Flankers are from notMNIST and Omniglot datasets.