[Reviews · NeurIPS 2017]

Reviewer 1



The paper study the effect of crowding, under particular conditions, on the object recognition performance of deep neural networks. The authors considered different DNN architectures, in particular Convnets and eccentricity-dependent networks, both with different pooling strategies. For the convnets they tried: (i) no pooling (60-54-48-42), (ii) progressive pooling (60-27-11-1), and (iii) at the end pooling (60-54-48-1) and trained with minibatches of 128 images and 32 feature channels. The eccentricity-dependent network constructs a multiscale representation where smaller sections are sample densely at a high resolution and larger sections are sampled with at a lower resolution. The configurations for this type of network were: (i) at the beginning (11-1-1-1-1), (ii) progressively (11-7-5-3-1), and (iii) at the end (11-11-11-11-1). Models were trained on even MNIST digits with/without flankers of different datasets and embedded in a dataset of places. From the results it can be seen that when the models are trained with isolated objects, the recognition rate falls when flankers are added to the image. This is specially true with flankers that are similar to the objects to be recognized. The eccentricity dependent network has better results when the object to recognize is in the center of the image. This is an interesting paper that is testing the robustness of DNNs and helps to explain some degradation in performance when different background objects are shown in testing images.

Reviewer 2



​Crowding is an effect in human vision, in which objects that can be recognized in isolation can no longer be recognized in the presence of nearby objects, even though there is no occlusion. This paper tries to answer the question whether deep neural network suffers from the crowding effect as well as the relationship of crowding to various types of configurations such as pooling, eccentricity, target-flanker similarity, flanker configurations etc. I think this paper is a good attempt of understand crowding of neural networks, but the execution of this work needs major improvements. ​ 1. Most of the result shown in this paper are well expected. For example, neural network could suffer from crowding, and training the network with the same type of flankers can improve the performance. 2. Some of the statements are not well supported by the experiments. For example, line 168 "thus in order for a model to be truly robust to all kinds of clutter, it needs to be trained with all possible target-flanker configurations". This is not well justified, since the experiments are conducted based on 120x and 240x spacing only. 3. The study is purely empirical and the settings studied are quite limited in their scope. For example, it is stated that "we investigate pooling in particular, because some computational models of crowding have suggested that feature integration may partly be the cause of this phenomenon in humans". However, neural networks are very different from human vision. The effect of crowding may be well based on other aspects of the network rather than the pooling operation. 4. I do not see utilities of the proposed method. Towards the end of this paper, "our results suggest that the eccentricity -dependent model coupled with a system for selecting eye fixation location would give benefit of low sample complexity in training". This is a conjecture rather than conclusion.

Reviewer 3



This paper studies if crowding, a visual effect suffered by human visual systems, happens to deep neural network as well. The paper systematically analyzes the performance difference when (1) clutter/flankers is present; (2) the similarity and proximity to the target; (3) when different architectures of the network is used. Pros: There are very few papers to study if various visual perceptual phenomenon exists in deep neural nets, or in vision algorithms in general. This paper studies the effect of crowding in DNN/DCNN image classification problem, and presents some interesting results which seems to suggest similar effect exists in DNN because of pooling layers merges nearby responses. And this is related to the theories of crowding in humans, which is also interesting. The paper also suggests what we should not do prematurely pooling if when designing architectures. In my opinion such papers should be encouraged. Cons: My main criticism to the paper is that it solely studies crowding in the context of image classification. However, if crowding is studied in the context of object detection, where the task is localize the object and recognize its category, the effect may be significantly lessened. For example, R-CNN proposes high saliency regions where the object might be and perform classification on that masked region. Because targets are usually centered in such proposed region and background clutters are excluded from the proposed region, the accuracy can potentially be much higher. After all, the extent to which crowding is present in DNN depends a lot on the chosen architecture. And the architecture in this paper is very primitive compare to what researchers consider state-of-the-art these days, and the accuracy of the MNIST tasks reported by the paper are way lower than what most researchers would expect from a practical system. For example, [1] performs digit OCR which has much more clutters but with very high accuracy. It is not obvious architectures like that also suffer from crowding. Suggestion: The paper is overall easy to follow. I feel the experimental setup can be more clear if some more example images (non-cropped, like the ones in the Fig1 of supplementary material). Overall, this paper has an interesting topic and is a nice read. The conclusions are not too strong because it uses simplistic architecture/datasets. But I think it is nonetheless this is a helpful paper to (re-)generate some interest on drawing relation between theories of human visual recognition and neural nets. [1] Goodfellow, Ian J., et al. "Multi-digit number recognition from street view imagery using deep convolutional neural networks." arXiv preprint arXiv:1312.6082 (2013).